# 3D Reconstruction of Wrist Bones from C-Arm Fluoroscopy Using Planar Markers

**DOI:** 10.3390/diagnostics13020330

**Published:** 2023-01-16

**Authors:** Pragyan Shrestha, Chun Xie, Hidehiko Shishido, Yuichi Yoshii, Itaru Kitahara

**Affiliations:** 1Graduate School of Science and Technology, Degree Programs in Systems and Information Engineering, Doctoral Program in Empowerment Informatics, University of Tsukuba, Tsukuba 305-8577, Japan; 2Center for Computational Sciences, University of Tsukuba, Tsukuba 305-8577, Japan; 3Department of Orthopedic Surgery, Tokyo Medical University Ibaraki Medical Center, Ami 300-0395, Japan

**Keywords:** three-dimensional data, tracking, computed tomography, fluoroscopy, preoperative plan, distal radius fracture

## Abstract

In orthopedic surgeries, such as osteotomy and osteosynthesis, an intraoperative 3D reconstruction of the bone would enable surgeons to quickly assess the fracture reduction procedure with preoperative planning. Scanners equipped with such functionality are often more expensive than a conventional C-arm fluoroscopy device. Moreover, a C-arm fluoroscopy device is commonly available in many orthopedic facilities. Based on the widespread use of such equipment, this paper proposes a method to reconstruct the 3D structure of bone with a conventional C-arm fluoroscopy device. We focus on wrist bones as the target of reconstruction in this research as this will facilitate a flexible imaging scheme. Planar markers are attached to the target object and are tracked in the fluoroscopic image for C-arm pose estimation. The initial calibration of the device is conducted using a checkerboard pattern. In general, reconstruction algorithms are sensitive to geometric calibration errors. To assess the practicality of the method for reconstruction, a simulation study demonstrating the effect of checkerboard thickness and spherical marker size on reconstruction quality was conducted.

## 1. Introduction

Radiographic imaging is a crucial technology in modern healthcare systems and medical diagnostics. Interventional radiologists use minimally invasive procedures guided by various modalities of medical imaging. Orthopedic surgeries, such as osteosynthesis and osteotomies, also benefit from intra-operative X-ray imaging [1,2]. Computed tomography (CT) scans prior to surgery are used to plan and simulate the fracture reduction process, while intra-operative X-ray images are used as guidance for positioning implants or cutting bones during surgery [3,4,5,6,7]. However, an apparent drawback with such an image-based navigation system is the loss of depth perception due to the transmission nature of X-ray images. To compensate for the loss of 3D information, a line of research focuses on registering preoperative 3D models to the intraoperative X-ray images [8]. While this serves as visual guidance during surgery, it would be beneficial to also have a registered 3D model of the current subject undergoing the surgery. This need could be fulfilled by mobile cone beam CT (CBCT) devices which have become more common in recent years [9]. These devices have a motorized iso-centric arm that can capture motion trajectory with high precision and reconstruct the volume inside the region of interest. However, many C-arm devices that are currently in practical use do not have tomographic capabilities since they are not designed for such use cases. Moreover, the initial cost of installing a mobile CBCT device is too high for rural sites. Therefore, a simple and easy-to-setup system for 3D reconstruction using a conventional C-arm fluoroscopy device would be beneficial in such cases.

In the field of computer vision, 3D reconstruction from multi-view images of a scene has been researched extensively [10,11]. In structure from motion [11], the problems of camera calibration and pose estimation are solved by decomposing the fundamental matrix obtained by point-to-point correspondences in multi-view images, which is followed by the triangulation step for initial reconstruction. Unfortunately, these methods cannot be applied to X-ray images since point correspondences cannot be established due to overlapping structures. In the field of CT reconstruction, techniques such as FDK- (Feldkamp, Davis, Kress) [12] and ART-based (Algebraic Reconstruction Technique) [13] methods are state-of-the-art techniques for tomographic reconstruction. These techniques require accurate geometry information beforehand. Therefore, accurate geometric calibration is required for applying these techniques in devices that are not designed for the use of these techniques. To circumvent the problem of accurate geometric calibration, some of the related works use a data-driven approach for reconstruction. In [14,15,16], the authors reconstructed the shape of a distal femur from multiple X-ray images using statistical shape models. These methods first require a statistical shape model, which is obtained by applying a principal component analysis to various CT models of similar structures. The obtained model is deformed so that its simulated projection is consistent with the acquired real projection. In [17], the author proposed a deep learning-based approach that uses a generative adversarial network to reconstruct CT models from biplanar X-rays. On the other hand, other works have focused either on C-arm calibration [18,19], reconstruction [20], or both given the initial geometry estimates [21].

Closely related to our work, Abella et al. [22] proposed a low-cost solution for tomographic reconstruction using a 3D scanner mounted on the C-arm. They identified three major issues that occur during tomographic reconstruction using a conventional C-arm device. First, the acquisition trajectory may deviate from the circular path and the mechanical stress causes each projection to have different acquisition parameters (i.e., source position, detector position, detector rotations). Second, these acquisition parameters are not repeatable for consecutive acquisitions. Third, the problem of limited angle tomography needs to be solved. To solve these issues, they used the calibration phantom developed by Cho et al. [23] for the geometric calibration of the device. Furthermore, the geometric errors originating from the non-repeatability of the C-arm trajectory were corrected using the information from the 3D scanner. An algorithm incorporating surface information was also introduced to account for the limited-angle tomography problem.

In this paper, an image-based framework for the tomographic reconstruction of wrist bones is proposed. A simulation study to evaluate its practicality was conducted. The proposed method assumes rotation of the target subject instead of the C-arm, which is possible for wrist bones. This way, the internal parameters, such as detector-to-source distance and detector rotations, remain constant between projections. For estimating the internal parameters of the C-arm device, a single checkerboard-based calibration was used, as demonstrated in [24]. A simple, easy-to-use, attachable calibration plane is proposed to estimate the C-arm pose without relying on external sensors. An open-source library “Tomographic Iterative GPU-based Reconstruction Toolbox” (TIGRE) [25] in MATLAB was used for reconstructing the volume. Equations were derived to convert results from the camera system to the tomography device system used in TIGRE.

## 2. Materials and Methods

Figure 1 shows the flow diagram of the proposed method. The C-arm calibration step is required before the acquisition of the subject to estimate the internal parameters such as the distance between the source to the detector and the piercing point (i.e., the point of intersection of the detector plane and the ray originating from the source, which is perpendicular to the detector). Pose estimation involves tracking the planar markers and solving for extrinsic parameters in the camera setup. The camera parameters are then converted into CBCT geometry for TIGRE and the simultaneous iterative reconstruction technique (SIRT) algorithm is applied for 20 iterations to reconstruct the volume.

### 2.1. C-Arm Calibration

A pinhole camera model was assumed for the image formation process in a C-arm device. A point in the device coordinate was projected into the detector plane according to Equation (1).
(1)x=KR[I |−t]X
where K is the intrinsic matrix, R is the orientation of the X-ray camera and t is its translation vector, as seen from the world coordinate. The intrinsic matrix K was modeled as follows.
(2)$$K=\left[\begin{array}{ccc} f_{x} & 0 & c_{x}\\ 0 & f_{y} & c_{y}\\ 0 & 0 & 1 \end{array}\right]$$
where $f_{x}$ and $f_{y}$ are the focal lengths in respective pixel units, and $c_{x}$ and $c_{y}$ are the pixel coordinates of the principal point (i.e., piercing point). We did not model distortions for the sake of simplicity, but they can be factored in easily using the polynomial distortion models. We adopted the calibration method from [24]. The design of the checkerboard pattern is shown in Figure 2. It was a 4 by 5 squares checkerboard containing 12 identifiable feature points. The feature points were localized and identified in the multi-view projection images of the checkerboard. Zhang’s camera calibration algorithm was applied to obtain the intrinsic parameters.

### 2.2. Pose Estimation

The planar markers were designed with five spherical markers that formed a parallelogram, as shown in Figure 3 (left). The reason for adopting such a pattern was two-fold. First, the 10 mm gap between the vertices P1 and P3 in the vertical axis ensured that the markers remained visible at the areas where the plane normal was close to perpendicular to the camera view direction. Second, the introduction of marker P5 helped to identify the line P1-P5-P2 in the projection image by using the fact that perspectivity preserves collinearity.

Marker tracking required localizing the marker points by applying the circular Hough transform to the edge image. Then, for each permutation of the markers, the collinearity condition was checked to identify the P1-P5-P2 line segment. After the identification of the line segment, the markers were assigned according to their y-position in the image, i.e., the marker with the lowest y-value was assigned as P1. The remaining two markers were also assigned according to their y-value.

We continued with decomposing the homography matrix H to obtain the camera pose for every view. An estimate of camera rotation and translation could be derived as follows:(3)$$H=\left[h_{1}{\;h}_{2}{\;h}_{3}\right]$$
(4)$$r_{1}=\lambda K^{-1}h_{1}$$
(5)$$r_{2}=\lambda K^{-1}h_{2}$$
(6)$$r_{3}=r_{1}\times r_{2}$$
(7)$$t=\lambda K^{-1}h_{3}$$
$$\lambda =\frac{1}{\left||K^{-1}h|\right|}$$

The rotation matrix was obtained by approximating the best rotation defined by matrix $\left[\begin{array}{ccc} r_{1} & r_{2} & r_{3} \end{array}\right]$. Further refinement of the external parameters was achieved by minimizing the reprojection error. The obtained rotation matrix and translation vectors were defined for the coordinate system shown in Figure 3 (right).

### 2.3. Reconstruction

The parameters estimated in the camera system needed to be converted to those used in TIGRE. The parameters involved are summarized in Table 1. The conversion was derived from the geometry observed in Figure 4. During pose estimation, projection images with mean reprojection errors larger than a given threshold (outliner data) were excluded because they degrade the reconstruction image quality. For reconstruction of the volume, the simultaneous iterative reconstruction technique (SIRT) in TIGRE was used with 20 iterations for all experiments.
(8)$$R^{T}=\left[\begin{array}{ccc} x & y & z \end{array}\right]\;$$
(9)$$t=-R^{T}*t$$
(10)d=(t·z)z−t
(11)DSO=||t−d||
(12)$$DSD=f_{x}*\alpha$$
(13)$$R_{\mathrm{source}}=\left[\begin{array}{ccc} -z & x & y \end{array}\right]$$
where R is the camera orientation matrix, t is the translation vector, d is the image offset vector, α is the size of the detector per pixel and $R_{\mathrm{source}}$ is the rotation matrix for computing Euler angles. The detector rotation and offsets are assumed to be zero in this conversion equations, even if is present in actual system. However, it does not not affect the reconstruction algorithm because the parameters obtained from the above calculation encodes such information (i.e., equivalent to redefining the coordinate system such that detector rotation and offsets became zero in each view independently).

## 3. Experiments and Results

### 3.1. Simulation Setup

A simulation environment was set up to investigate the effect of different calibration board thicknesses and spherical marker sizes on image feature points as well as image quality. TIGRE for MATLAB was used for simulating the X-ray generation, as well as reconstruction algorithms. We used the internal parameters of the SIEMENS CIOS Select tool for simulating the C-arm. The common configurations used for simulated X-ray generation are shown in Table 2.

A 3D model of the checkerboard with a resolution of 0.25 mm/vx and a size of 100 mm in all three dimensions was built with Blender and converted into voxel data with the interior filled with a constant value. Figure 5 shows the 3D model along with an example of an X-ray image in a particular pose. A CT scan of the wrist phantom with a resolution of 0.25 mm/vx in all three dimensions and a size of 151 mm for width and depth, measuring 161 mm in height, was used for simulating X-rays. Figure 6 shows the volume rendering of the CT model along with an example X-ray image at a 0-degree rotation. The planar markers were attached to the posterior region of the phantom by editing the voxel values. During pose estimation, the threshold value for rejecting outliers (i.e., projection images and geometries) based on the reprojection error was set to 50 pixels. This threshold was selected manually by inspecting the distribution of the reprojection error so that at least 80% of the original images were retained, while preventing sparse view-related artifacts from dominating the reconstruction error. Evaluation of the reconstructed volume was performed using the result of the SIRT reconstructed volume with ground truth acquisition geometry for calibration board and marker size. Figure 7 shows the volume rendering of the SIRT reconstructed volume with ground truth geometry with a 1 mm board size and 2 mm marker size.

### 3.2. Effect of Board Thicknesses and Marker Sizes on Reconstructed Image Quality

The proposed method for calibration uses a checkerboard pattern to identify feature points. However, in practice, radio-opaque lead plates used for black squares are quite thick. This will create ambiguities in the detected feature points. An example of such ambiguities is shown in Figure 8. For checkerboard patterns, the world feature points were in the corner of the square, halfway through the board’s thickness. For spherical planar markers, the world feature points were considered to be the centers of each sphere. However, due to perspectivity, its projection in the image plane deviated from the detected center of the ellipse. Due to these ambiguities in the image points, the resulting reconstruction quality may degrade depending on the chosen sizes of these materials. Therefore, we evaluated the proposed method with four calibration board sizes ranging from 1 mm to 5 mm in thickness and five spherical marker sizes ranging from 2 mm to 6 mm in diameter, which are the commonly available sizes. A 1 mm spherical marker was eliminated due to the resolution of the X-ray simulating volume.

Table 3 shows the structural similarity index measure (SSIM) compared to ground truth reconstruction for each case. The best case of 0.9441 was achieved with 3 mm board thickness and a marker size of 2 mm. The worst case of 0.8982 was obtained with 5 mm board thickness and a marker size of 6 mm. Detection errors for planar markers and the feature points in the calibration board are shown in Figure 9. Although a total of 180 projection images were captured around a 180 degrees angle of rotation, projection images where the planar markers were projected in a lateral view were prone to pose estimation errors, as well as wrong marker identification. These outlier images were rejected using simple filtering of the reprojection error. Figure 10 shows the number of projection images that were used (i.e., after the filtering) for reconstruction for each marker size and board thickness. An example of the volume rendering and volume slices of the reconstruction with a 3 mm thickness calibration board and 2 mm planar marker (best case) is shown in Figure 11. A similar example with a 5 mm thickness calibration board and 6 mm planar marker (worst case) is shown in Figure 12.

## 4. Discussion

From Table 3, we can observe that the reconstructed image quality decreased as the marker size increased. Although the image quality degraded when comparing reconstruction results of a sample with 1 mm board thickness with one with a 5 mm board thickness, the effect was not as significant as those obtained when changing the marker size. This suggests that the geometrical calibration of internal parameters obtained with checkerboard calibration can be used for image reconstruction. As opposed to the calibration phantom in [23], the checkerboard pattern was easier to prepare, and off-the-shelf software is readily available for this kind of planar calibration.

On the other hand, the steady drop in the number of used images shown in Figure 10 was due to the larger detection error in larger marker sizes amplifying the reprojection error, as can be observed from Figure 11, to above the threshold of 50 pixels for outlier images. The effect of this variation in the number of images used can be observed in the reconstructed image in Figure 11 and Figure 12. The slice images in both cases had streak artifacts that were a result of insufficient projection angles. The slice image in Figure 12, however, contained metal artifacts resulting from a larger marker size as well. This suggests that the size of the spherical marker in our proposed method was crucial in obtaining a good reconstruction quality. From Table 3, it can be seen that approximately a 2% increase in SSIM could be confirmed when compared to using a 3 mm marker size. Thus, it can be concluded that the proposed method leads to better results when a 2 mm marker size is selected.

Since the proposed method cannot process projection images in which the projections of the markers are close to being co-linear, there is an inherent limitation to the image quality which can be seen in Figure 11, which shows the reconstructed image with the configuration that led to the best SSIM. Additionally, our method assumes constant intrinsic parameters throughout the acquisition. This limits its applicability to structures that cannot be rotated easily such as pelvis.

Due to the recent advances in preoperative 3D programs for various procedures in orthopedic surgery [3,26,27], it is necessary to establish a method for comparing 3D models of a preoperative plan to 3D models during surgery. The method developed here has the potential to facilitate the comparison of intraoperative 3D models with 3D images of preoperative planning. In previous studies, the reduction accuracy of 3D preoperative planning was moderate. This is because it is difficult to visualize the 3D model of the reduction shape during surgery. Thus, this method will be useful in the clinical field with comparisons to a preoperative 3D model. The advantage of this method is that there is no need to install a new system in the operating room. Furthermore, the proposed method does not require the installation of external devices, which makes it easier to adapt to many existing systems.

In conclusion, this paper proposed an image-based reconstruction framework using checkerboard calibration and rotation of the target subject with attached plane markers. It was found that the image quality of the reconstruction depended heavily on the size of the planar markers used. Smaller markers could be detected with low image point error, resulting in an increased number of reliable projection images and geometries that were used for reconstruction. The effect of calibration board thickness is not as significant as that of the planar marker size, but it should be kept below 4 mm for optimal results. Further improvements can be achieved by installing another set of planar markers attached perpendicular to the current one and by solving the camera pose estimation problem with Perspective-n-Point [28], so that lateral views can also be incorporated into the reconstruction pipeline. This is possible due to the cylindrical nature of wrist bones.

## Figures and Tables

**Figure 1 diagnostics-13-00330-f001:**
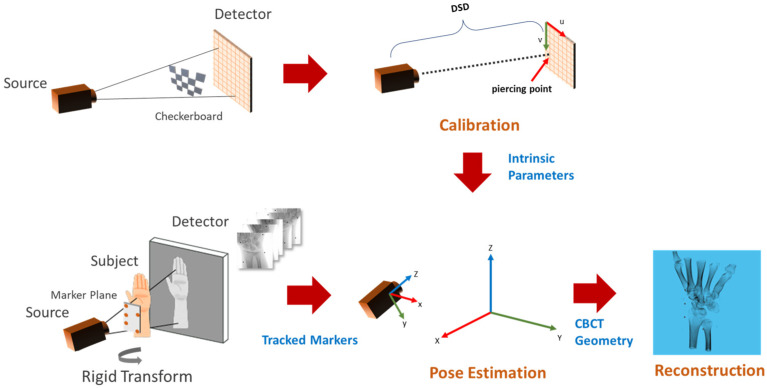
Overview of the calibration and reconstruction pipeline. The C-arm is calibrated using multiple captures (**top left**) of the calibration checkerboard to obtain intrinsic parameters (**top-center**). A planar marker board is attached to the subject before X-ray acquisition (**bottom left**). Markers in the acquired multi-view images are tracked for pose estimation (**bottom center**), followed by the reconstruction algorithm (**bottom right**).

**Figure 2 diagnostics-13-00330-f002:**
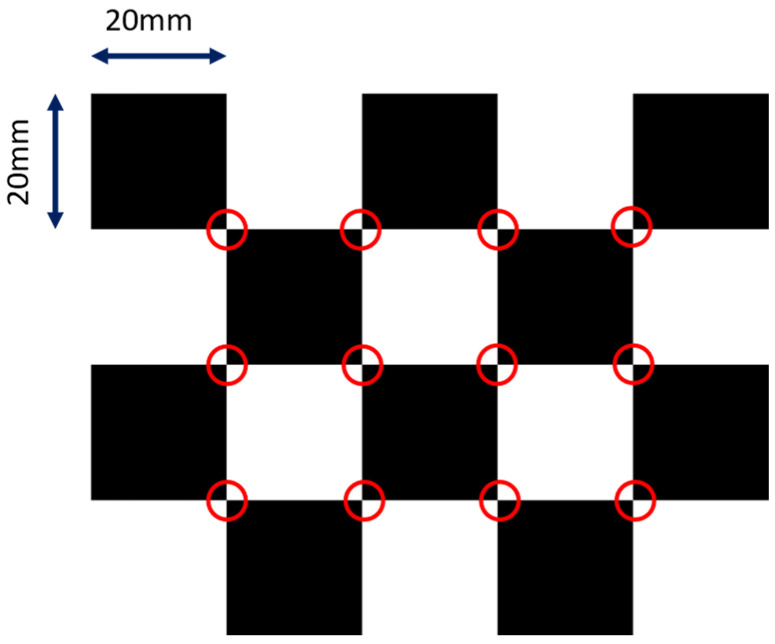
The layout of the checkerboard. The detectable image feature points are marked in red circles.

**Figure 3 diagnostics-13-00330-f003:**
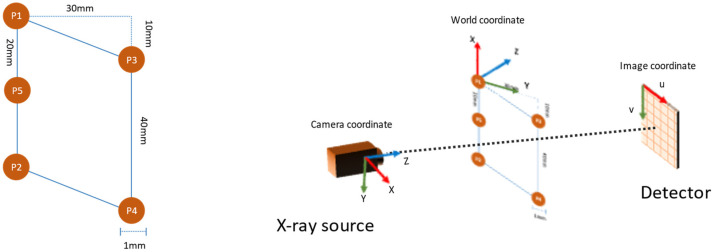
The layout of the planar markers (**left**) and the coordinate system used in pose estimation (**right**).

**Figure 4 diagnostics-13-00330-f004:**
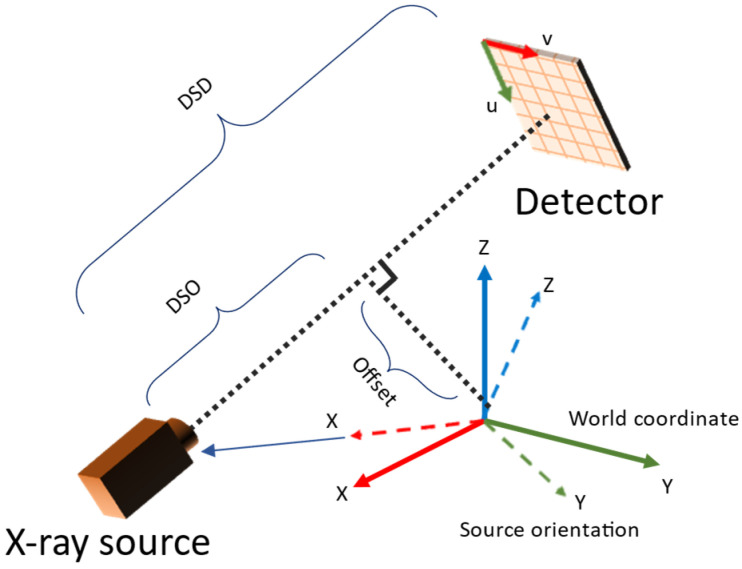
Relation between TIGRE geometry parameters and camera system setup.

**Figure 5 diagnostics-13-00330-f005:**
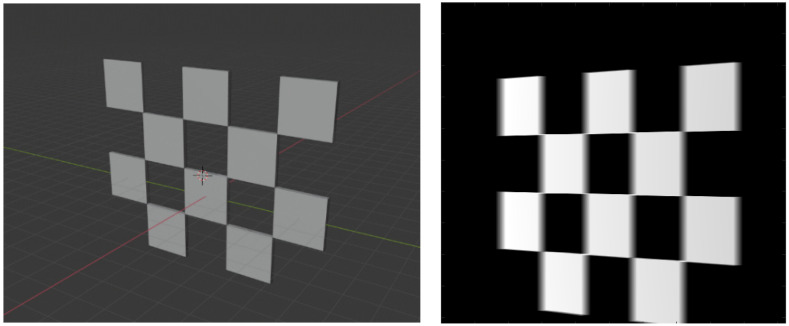
A 3D view of calibration board in Blender (**left**). A projection of the board (**right**).

**Figure 6 diagnostics-13-00330-f006:**
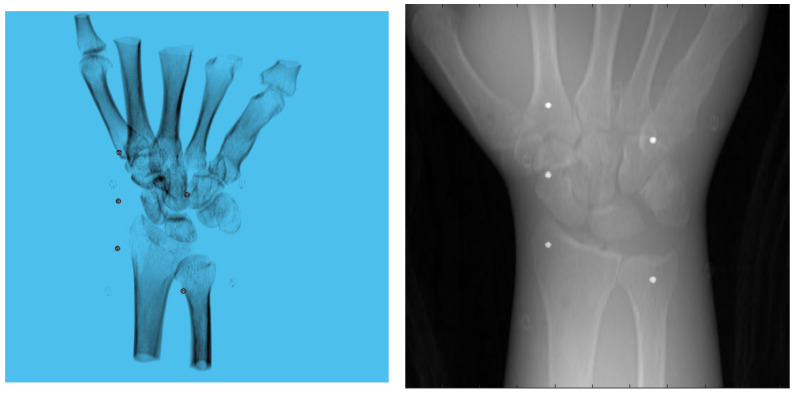
Volume rendering of the CT scan of wrist phantom (**left**). A projection of the volume (**right**).

**Figure 7 diagnostics-13-00330-f007:**
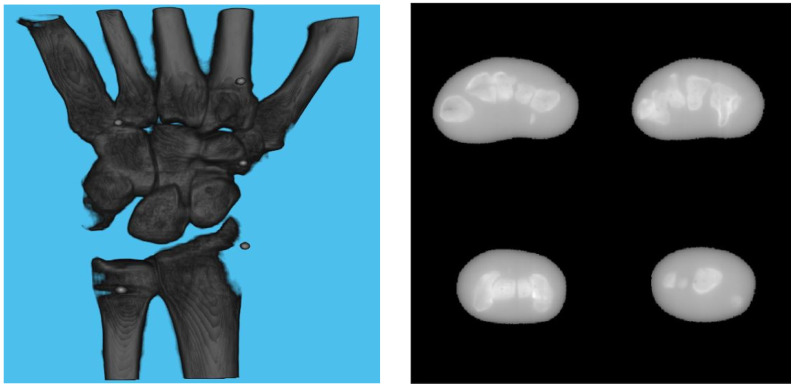
Volume rendering of the reconstructed volume using ground truth projection and geometries (**left**). Four fixed slices (**top left**: meta carpals, **top right**: carpals, **bottom right**: carpals, **bottom left**: radius and ulna) of the volume (**right**).

**Figure 8 diagnostics-13-00330-f008:**
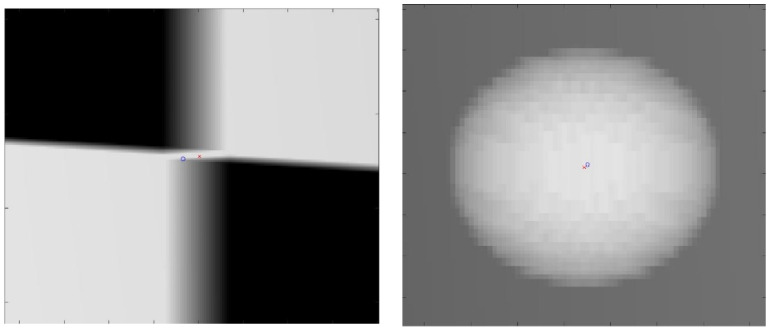
Close-up image of the projection of the calibration checkerboard (**left**). Close-up image of the projection of spherical marker (**right**). The red cross is the ground truth image point; the blue circle is the detected image point.

**Figure 9 diagnostics-13-00330-f009:**
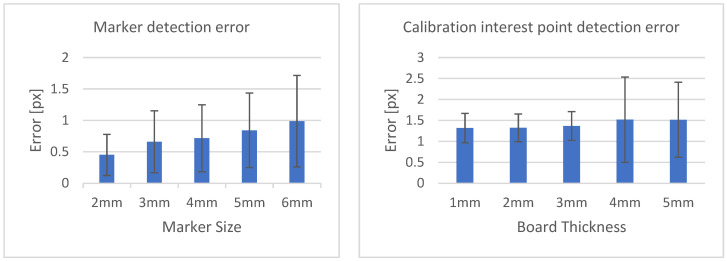
Marker detection error according to marker size (**left**). Calibration interest point detection error according to board thickness (**right**).

**Figure 10 diagnostics-13-00330-f010:**
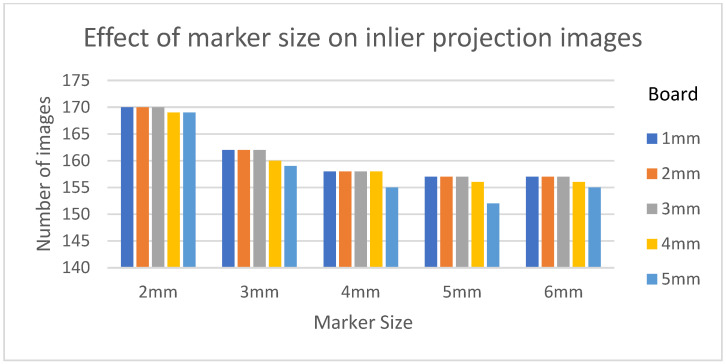
The number of inlier projection images as the size of the planar marker and calibration board thickness changes.

**Figure 11 diagnostics-13-00330-f011:**
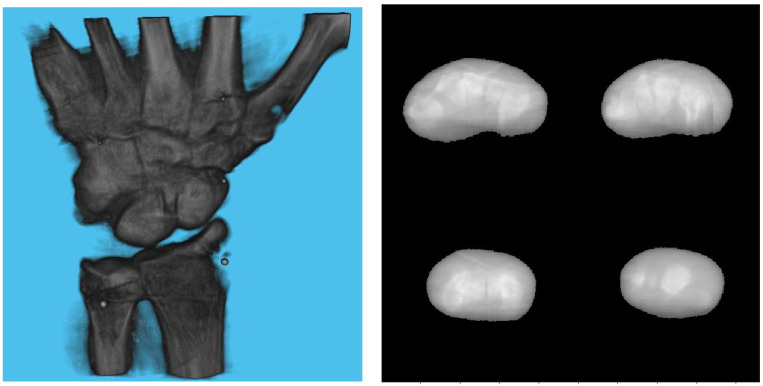
Volume rendering of the reconstructed volume using a 3 mm calibration board and 2 mm marker (**left**). Four fixed slices (**top left**: metacarpals, **top right**: carpals, **bottom right**: carpals, **bottom left**: radius and ulna) of the volume (**right**).

**Figure 12 diagnostics-13-00330-f012:**
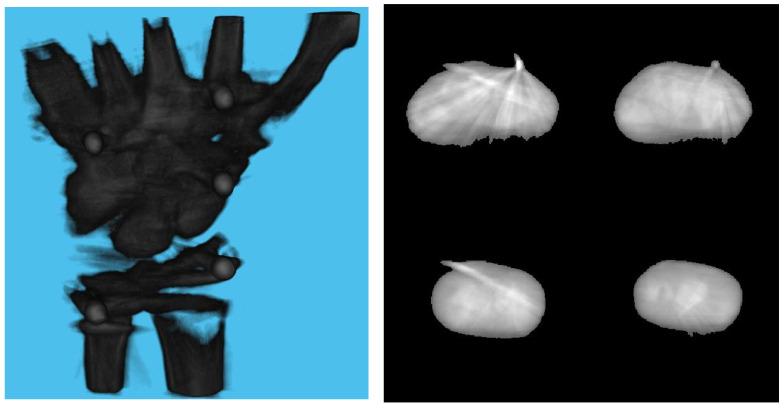
Volume rendering of the reconstructed volume using a 5 mm calibration board and 6 mm marker (**left**). Four fixed slices (**top left**: metacarpals, **top right**: carpals, **bottom right**: carpals, **bottom left**: radius and ulna) of the volume (**right**).

**Table 1 diagnostics-13-00330-t001:** List of parameter names used in TIGRE along with their descriptions.

Parameter Name	Description
DSD	Distance between the X-ray source and detector plane
DSO	Distance between the X-ray source and world origin
offOrigin	Offset applied to the volume in world origin
offDetector	Offset applied to the detector
angles	The angle of X-ray source from world origin in ZYZ convention
nVoxel	Number of voxels in the volume
dVoxel	Size of one voxel in the world coordinate
sVoxel	Total size occupied by the volume in the world coordinate
nDetector	Number of pixels in the detector panel
sDetector	The total size of the detector panel in the world coordinate

**Table 2 diagnostics-13-00330-t002:** Values of some common parameters used in the simulation.

Parameter Name	Value
DSD	780 mm
DSO	390 mm
offOrigin	(0, 0, 0)
offDetector	(0, 0)
Angles	180 samples between 0 and 180 degrees
nDetector	(1024 px, 1024 px)
sDetector	210 mm
Calibration Board	
nVoxel [vx]	(500, 500, 500)
dVoxel [mm/vx]	(0.25, 0.25, 0.25)
sVoxel [mm]	(100, 100, 100)
CT Volume for Simulated X-ray	
nVoxel [vx]	(604, 604, 644)
dVoxel [mm/vx]	(0.25, 0.25, 0.25)
sVoxel [mm]	(151, 151, 161)
Reconstruction Volume	
nVoxel [vx]	(300, 300, 300)
dVoxel [mm/vx]	(0.5, 0.5, 0.5)
sVoxel [mm]	(150, 150, 150)

**Table 3 diagnostics-13-00330-t003:** SSIM of the reconstructed volume with ground truth reconstructed volume for each configuration pair. For each marker size, the board thickness that resulted in the best SSIM are marked in bold.

Board Thickness/Marker Size	2 mm	3 mm	4 mm	5 mm	6 mm
1 mm	0.9435	0.9205	0.9235	0.9120	**0.9114**
2 mm	0.9437	0.9206	0.9236	0.9122	0.9111
3 mm	**0.9441**	0.9206	**0.9237**	**0.9125**	0.9107
4 mm	0.9426	**0.9225**	0.9236	0.9116	0.9043
5 mm	0.9352	0.9218	0.9221	0.9107	0.8982

## Data Availability

The datasets analyzed during the present research are available from the corresponding author upon reasonable request.

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
