# Peer review of "3D Reconstruction of Wrist Bones from C-Arm Fluoroscopy Using Planar Markers"

_diagnostics, 2023, doi:10.3390/diagnostics13020330_

Round 1

Reviewer 1 Report

The manuscript presented to me for review is a pioneering study, focused on creating a pictorial reconstruction of the wrist bone using CT. A novelty is the assumption of the possibility of rotation of the examined object, and not the arm C. According to the authors, such a possibility is provided by the wrist bones. This procedure guarantees the stability of internal calibration parameters such as the detector/radiation source distance and the rotation of the detector. In addition, the use of the chessboard as a calibration table for the position of the arm makes the process independent of external sensors.

This work is part of a series of publications aimed at creating precise mechanisms, using CT and mathematical modeling, allowing for more accurate bone reconstruction, which is of key importance for orthopedics. However, the pioneer character of this work does not absolve the authors from its properly discussing in accordance with the applicable rules. This form of discussion is unacceptable

Author Response

Thank you for reviewing our paper. We revised the manuscript and corrected errors that were pointed out in the review comments. Regarding the discussion section, We have added two main conclusions that could be drawn from the obtained results from a technical perspective (Page 10, Line 810 and Line 821). A paragraph discussing the clinical relevance of the method (Page 10, Line 833) and limitations (Page 10, Line 827) of the proposed method has also been added. 
The revised part is as follows. 
---------------------------- 
Page 10, Line 810 

This suggests that the geometrical calibration of internal parameters obtained with checkerboard calibration can be used for image reconstruction. As opposed to the calibration phantom in [23], the checkerboard pattern is easier to prepare, and off-the-shelf software is readily available for this kind of planar calibration. 

Page 10, Line 821 

This suggests that the size of the spherical marker in our proposed method is crucial to obtaining a good reconstruction quality. From Table 3, it can be seen that approximately a 2% increase in SSIM can be confirmed when compared to using a 3 mm marker size. Thus, it can be concluded that the proposed method leads to better results when a 2 mm marker size is selected. 

Page 10, Line 833 

Due to the recent advances in preoperative 3D programs for various procedures in orthopedic surgery [3, 26, 27], it is necessary to establish a method for comparing 3D models of a preoperative plan to 3D models during surgery. The method developed here has the potential to facilitate the comparison of intraoperative 3D models with 3D images of preoperative planning. In previous studies, the reduction accuracy of 3D preoperative planning was moderate. This is because it is difficult to visualize the 3D model of the reduction shape during surgery. Thus, this method will be useful in the clinical field with comparisons to a preoperative 3D model. The advantage of this method is that there is no need to install a new system in the operating room. Furthermore, the proposed method does not require the installation of external devices, which makes it easier to adapt to many existing systems. 

Page 10, Line 827 

Since the proposed method cannot process projection images in which the projections of the markers are close to being co-linear, there is an inherent limitation to the image quality which can be seen in Figure 11, which shows the reconstructed image with the configuration that led to the best SSIM. Also, our method assumes constant intrinsic parameters throughout the acquisition. This limits its applicability to structures that cannot be rotated easily such as pelvis. 
----------------------------

Reviewer 2 Report

This manuscript is based on C-arm fluoroscopy images to assess the quality of 3D reconstruction of wrist bone. The study was interesting and well organized, but the manuscript has some written problems with multiple grammatical errors that need addressing.

Generally, the topic cannot clearly present the real purpose and results of the experiment in this article. It is recommended to make appropriate adjustments. Moreover, the devices, machines and softwares what used in this experiment should indicate the brand, model, and origin details.

The following questions and comments may help for the author to further improve the manuscript quality.

Lines 95~97, What is the purpose of the source in the lower left corner of Figure 1 being marked in green (others are brown)? Please explain in detail in the legend for readers to understand.

Lines 109~110 and 114, “The design of checkerboard pattern is shown in Figure 2. It is a 4 by 5 squares checkerboard containing 6 identifiable feature points.” and “Figure 2. Layout of the checkerboard. Red circles are the image feature points.” This article describes that checkerborder has six image feature points, but the red circle in Figure 2 has 12 image feature points. Could you explain it?

Line 177 Table 2, sVoxel [mm] in CT Volume for Simulated X-Ray (151, 151, 161), Please explain why the last number is different from the first two? This is different from the case for other parameters.

Lines 182~183, “During pose estimation, the threshold value for rejecting outliers (i.e., projection images and geometries) based on the reprojection error was set to 50 pixels.” What is the reason for setting the threshold at 50 pixels? Is there any literature to support this assertion?

Lines 187~194, Figures 5~7 only have pictures and legends, but they are not marked with corresponding statement positions in the main text.

Line 194, “Four slices in top to bottom manner of the volume” Where are these four slices, and where are they located? Is there a corresponding marker point?

Line 207, What is the full name of the acronym SSIM? Structural similarity index? Please check whether the full text abbreviates provide the full name when it appears for the first time.

Line 241, Discussion section, It is too brief, and repeated explanations and summaries are made on the results, which cannot reflect the purpose and value of this experiment. Moreover, the discussion section only discusses the accuracy differences between checkerboards and planar markers of different sizes, which seems to be unable to convince us that this method can be applied to the reconstruction accuracy of multiple complex three-dimensional wrist structures.

Author Response

Your comments:
I do not agree PFP is effective as less than 50% responds to the treatment, however effective for the group of patients who responds!

Authors' response:
This article discusses predictive factors for response to platelet rich plasma injections in knee osteoarthritis, that are currently unknown, in order to better target patients. There are many publications supporting the efficacy of PRP but none on the actual predictive factors of response. These results provide real information for our clinical practices.

Comments from previous reviewers:

1. I'm sorry to write that such works do not merit publication in special issue dedicated to biomarkers of osteoarthritis. is has no scientific merit other than: "we injected 210 knees , but only heel to buttock and ongoing physiotherapy improved.

2. It is an observational study without case control. The study is based on routinely constituted medical files, with consequently an insufficient level of standardization for data collection. The results are inconclusive and consequently do not help the medical community to decide if PRP can be useful for their patients.

Response to Reviewer 2 Comments

Thank you for reviewing our paper, we have revised the manuscript to address the questions and comments presented. We have numbered the questions/comments and linked them in the comments to corresponding sections of the manuscript.

R2-1 
The topic cannot clearly present the real purpose and results of the experiment in this article. It is recommended to make appropriate adjustments. The purpose of the experiment has been elaborated in Page 8, Line 601. The revised part is as follows. 
---------------------------- 
The proposed method for calibration uses a checkerboard pattern to identify feature points. However, in practice, radio-opaque lead plates used for black squares are quite thick. This will create ambiguities in the detected feature points. An example of such ambiguities is shown in Figure 8. For checkerboard patterns, the world feature points are in the corner of the square, halfway through the board’s thickness. For spherical planar markers, the world feature points are considered to be the centers of each sphere. However, due to perspectivity, its projection in the image plane deviates from the detected center of the ellipse. Due to these ambiguities in the image points, the resulting reconstruction quality may degrade depending on the chosen sizes of these materials. Therefore, we evaluated the proposed method with four calibration board sizes ranging from 1 mm to 5 mm in thickness and five spherical marker sizes ranging from 2 mm to 6 mm in diameter, which are the commonly available sizes. A 1 mm spherical marker was eliminated due to the resolution of the X-Ray simulating volume. 
—---------------------- 
R2-2 
The devices, machines and softwares what used in this experiment should indicate the brand, model, and origin details. This has been addressed in Page 6, 525. The revised part is as follows.
 —------------------------ 
We used the internal parameters of the SIEMENS CIOS Select tool for simulating the C-arm. 
—------------------------
R2-3 
Lines 95~97, What is the purpose of the source in the lower left corner of Figure 1 being marked in green (others are brown)? Please explain in detail in the legend for readers to understand. The purpose of the source is to emit X-ray toward the detector in this case. As pointed out, the color is united to brown for all the sources. Details of the diagram have been added in the legend of Figure 1 (Page 3., Line 345). The revised part is as follows. 
—------------------------ 
Figure 1. Overview of the calibration and reconstruction pipeline. The C-arm is calibrated using multiple captures (top left) of the calibration checkerboard to obtain intrinsic parameters (top-center). A planar marker board is attached to the subject before X-ray acquisition (bottom left). Markers in the acquired multi-view images are tracked for pose estimation (bottom center), followed by the reconstruction algorithm (bottom right).
 —------------------------- 
R2-4 
Lines 109~110 and 114, “The design of checkerboard pattern is shown in Figure 2. It is a 4 by 5 squares checkerboard containing 6 identifiable feature points.” and “Figure 2. Layout of the checkerboard. Red circles are the image feature points.” This article describes that checkerborder has six image feature points, but the red circle in Figure 2 has 12 image feature points. Could you explain it? Thank you for pointing out the mistake. This has been corrected to "12 identifiable feature points" (Page 4, Line 393). The revised part is as follows. 
—------------------------------ 
The design of the checkerboard pattern is shown in Figure 2. It is a 4 by 5 squares checkerboard containing 12 identifiable feature points. 
—----------------------- 
R2-5 
Line 177 Table 2, sVoxel [mm] in CT Volume for Simulated X-Ray (151, 151, 161), Please explain why the last number is different from the first two? This is different from the case for other parameters. The last number in (151, 151, 161) denotes height of the volume. These are the dimensions of the original CT volume obtained by scanning the wrist phantom, which is why they are different from other parameters that we set manually. A brief explanation of ct scan used in this experiment has been added in Page 6, Line 532. The revised part is as follows. 
—--------------------------

Figure 5 shows the 3D model along with an example of an X-ray image in a particular pose. A CT scan of the wrist phantom with a resolution of 0.25 mm/vx in all three dimensions and a size of 151 mm for width and depth, measuring 161 mm in height, was used for simulating X-Rays. 
—------------------------- 
R2-6 
Lines 182~183, “During pose estimation, the threshold value for rejecting outliers (i.e., projection images and geometries) based on the reprojection error was set to 50 pixels.” What is the reason for setting the threshold at 50 pixels? Is there any literature to support this assertion? This threshold was chosen manually by inspecting the distribution of the reprojection error. The objective was to reject any trivial outliers that arise from the misidentification of markers near perpendicular views as well as keep at least 80% of the original images (which is around 150 images) so as to abstain sparse view-related artifacts from dominating the reconstruction error. An explanation has been added in Page 7, 554. The revised part is as follows. 
—------------------------------------- 
This threshold was selected manually by inspecting the distribution of the reprojection error so that at least 80% of the original images were retained, while preventing sparse view-related artifacts from dominating the reconstruction error. 
—----------------------------------- 
R2-7 
Lines 187~194, Figures 5~7 only have pictures and legends, but they are not marked with corresponding statement positions in the main text. Thank you for pointing out the mistake, corresponding statements have been added in Page 6, Line 530. The revised part is as follows. 
—------------------------------------- 
Figure 5 shows the 3D model along with an example of an X-ray image in a particular pose. A CT scan of the wrist phantom with a resolution of 0.25 mm/vx in all three dimensions and a size of 151 mm for width and depth, measuring 161 mm in height, was used for simulating X-Rays . Figure 6 shows the volume rendering of the CT model along with an example X-ray image at a 0-degree rotation. The planar markers were attached to the posterior region of the phantom by editing the voxel values. During pose estimation, the threshold value for rejecting outliers (i.e., projection images and geometries) based on the reprojection error was set to 50 pixels. This threshold was selected manually by inspecting the distribution of the reprojection error so that at least 80% of the original images were retained, while preventing sparse view-related artifacts from dominating the reconstruction error. Evaluation of the reconstructed volume was performed using the result of the SIRT reconstructed volume with ground truth acquisition geometry for calibration board and marker size. Figure 7 shows the volume rendering of the SIRT reconstructed volume with ground truth geometry with a 1 mm board size and 2 mm marker size.

—------------------------------------- 
R2-8 
Line 194, “Four slices in top to bottom manner of the volume” Where are these four slices, and where are they located? Is there a corresponding marker point? These are arbitrarily fixed cross-sections, one slicing metacarpals, two for carpals, and one for radius and ulna. These slice points are kept the same for all the following results. Corresponding captions have been updated to reflect this in Page 10, Line 799 and 803. The revised part is as follows. 
—---------------------------------------- 
Figure 11. Volume rendering of the reconstructed volume using a 3 mm calibration board and 2 mm marker (left). Four fixed slices (top left: metacarpals, top right: carpals, bottom right: carpals, bottom left: radius and ulna) of the volume (right).
 —-------------------------------------- 
R2-9 
Line 207, What is the full name of the acronym SSIM? Structural similarity index? Please check whether the full text abbreviates provide the full name when it appears for the first time. Thank you for pointing it out. Yes, It is structural similarity index. A full text abbreviation had been added in Page 8, Line 614. The revised part is as follows. 
—-------------------------------------- 
Table 3 shows the structural similarity index measure (SSIM) compared to ground truth reconstruction for each case. 
—------------------------------------ 
R2-10 
Line 241, Discussion section, It is too brief, and repeated explanations and summaries are made on the results, which cannot reflect the purpose and value of this experiment. Moreover, the discussion section only discusses the accuracy differences between checkerboards and planar markers of different sizes, which seems to be unable to convince us that this method can be applied to the reconstruction accuracy of multiple complex three-dimensional wrist structures. Thank you for pointing it out. We have added two main conclusions that could be drawn from the obtained results from a technical perspective (Page 10, Line 810 and Line 821). A paragraph discussing the clinical relevance of the method (Page 10, Line 833) and limitations (Page 10, Line 827) of the proposed method has also been added. The revised part is as follows. 
---------------------------- 
Page 10, Line 810 
This suggests that the geometrical calibration of internal parameters obtained with checkerboard calibration can be used for image reconstruction. As opposed to the calibration phantom in [23], the checkerboard pattern is easier to prepare, and off-the-shelf software is readily available for this kind of planar calibration. 

Page 10, Line 821

This suggests that the size of the spherical marker in our proposed method is crucial to obtaining a good reconstruction quality. From Table 3, it can be seen that approximately a 2% increase in SSIM can be confirmed when compared to using a 3 mm marker size. Thus, it can be concluded that the proposed method leads to better results when a 2 mm marker size is selected. 

Page 10, Line 833 

Due to the recent advances in preoperative 3D programs for various procedures in orthopedic surgery [3, 26, 27], it is necessary to establish a method for comparing 3D models of a preoperative plan to 3D models during surgery. The method developed here has the potential to facilitate the comparison of intraoperative 3D models with 3D images of preoperative planning. In previous studies, the reduction accuracy of 3D preoperative planning was moderate. This is because it is difficult to visualize the 3D model of the reduction shape during surgery. Thus, this method will be useful in the clinical field with comparisons to a preoperative 3D model. The advantage of this method is that there is no need to install a new system in the operating room. Furthermore, the proposed method does not require the installation of external devices, which makes it easier to adapt to many existing systems. 

Page 10, Line 827 

Since the proposed method cannot process projection images in which the projections of the markers are close to being co-linear, there is an inherent limitation to the image quality which can be seen in Figure 11, which shows the reconstructed image with the configuration that led to the best SSIM. Also, our method assumes constant intrinsic parameters throughout the acquisition. This limits its applicability to structures that cannot be rotated easily such as pelvis. 
----------------------------

Round 2

Reviewer 2 Report

All the questions have been well addressed. I would suggest accepting this manuscript to publish.